# Learning-Augmented Data Stream Algorithms

**Tanqiu Jiang**
Department of Electrical and Computer Engineering
Lehigh University
Bethlehem, PA 18015, USA
taj320@lehigh.edu

**Yi Li**
School of Physical and Mathematical Sciences
Nanyang Technological University
Singapore 637371
yili@ntu.edu.sg

**Honghao Lin**
Zhiyuan College
Shanghai Jiao Tong University
Shanghai, China 200240
honghao_lin@sjtu.edu.cn

**Yisong Ruan**
Department of Software engineering
Xiamen University
Xiamen, Fujian, China 361000
24320152202802@stu.edu.xmu.cn

**David P. Woodruff**
Computer Science Department
Carnegie Mellon University
Pittsburgh, PA 15213, USA
dwoodruf@cs.cmu.edu

## Abstract

The data stream model is a fundamental model for processing massive data sets with limited memory and fast processing time. Recently Hsu et al. (2019) incorporated machine learning techniques into the data stream model in order to learn relevant patterns in the input data. Such techniques led to the training of an oracle to predict item frequencies in the streaming model. In this paper we explore the full power of such an oracle, showing that it can be applied to a wide array of problems in data streams, sometimes resulting in the first optimal bounds for such problems, and sometimes bypassing known lower bounds without such an oracle. We apply the oracle to counting distinct elements on the difference of streams, estimating frequency moments, estimating cascaded aggregates, and estimating moments of geometric data streams. For the distinct elements problem, we bypass the known lower bound and obtain a new space-optimal algorithm. For estimating the $p$-th frequency moment for $0 < p < 2$ we obtain the first algorithms with optimal space and update time. For estimating the $p$-th frequency moment for $p > 2$ we obtain a quadratic savings in memory, bypassing known lower bounds without an oracle. We empirically validate our results, demonstrating also our improvements in practice.

## 1 Introduction

Processing data streams has been an active research field in the past two decades. This is motivated by the increasingly common scenario where the size of the data far exceeds the size of the available storage, and the only feasible access to the data is to make a single or a few passes over the data. This situation occurs, for example, in network applications with high-speed packets being sent through routers with limited resources. Other examples include processing internet search logs, sensor networks, and scientific data streams. For an introduction to data streams, see, e.g., Muthukrishnan (2005).

Formally, in the data stream model, we assume there is an underlying *frequency vector* $x \in \mathbb{Z}^n$, initialized to $0^n$, which evolves throughout the course of a stream. The stream consists of updates of the form $(i, w)$, meaning $x_i \leftarrow x_i + w$. Suppose $M$ is such that $\|x\|_\infty = \max_{i \in \{1,...,n\}} |x_i| \leq M$

throughout the stream. At the end of the stream, we are asked to approximate $f(x)$ for a function $f : \mathbb{Z}^n \to \mathbb{R}$. Because of the sheer size of the stream, most algorithms are approximate and randomized. Typically, such algorithms output an estimate $Z$ for which $\Pr\{(1 - \epsilon)f(x) \leq Z \leq (1 + \epsilon)f(x)\} \geq 2/3$, where the probability is over the randomness used by the algorithm, and not of the input stream, which may be a worst-case stream. The success probability $2/3$ can be replaced with any constant greater than $1/2$ by independently repeating the algorithm a constant number of times and taking the median estimate. The space complexity of the algorithm is measured in bits and the goal is to use much less than the trivial $n$ bits of space required to store $x$.

A common difficulty in designing algorithms in the data stream model is that the coordinates $i$ for which $|x_i|$ is large, i.e., the *heavy hitters*, are not known in advance. Such coordinates usually require very accurate estimates in order to achieve an accurate final estimate of the function value. A number of techniques have emerged for identifying such coordinates, such as the Count-Min and CountSketch. These techniques are often used at multiple scales by subsampling the input coordinates and finding heavy items in the subsample, and have found applications to estimating frequency moments, cascaded aggregates, earthmover distance, and empirical entropy, among other statistics (Indyk & Woodruff, 2005; Chakrabarti et al., 2006; Jayram & Woodruff, 2009; Andoni et al., 2009; McGregor et al., 2016).

In many applications, however, the worst-case input is rarely seen; in fact, there is usually a pattern inside the underlying vector. For instance, in network traffic monitoring, the large or "elephant" flows fluctuate on a daily basis, but the actual source and destination servers of those flows do not vary much. In our formulation, this implies the value $x_i$ fluctuates while whether or not $x_i$ is a heavy coordinate remains stable. Motivated by this as well as related work on learning oracles for algorithm design (Gupta & Roughgarden, 2016; Dick et al., 2017; Balcan et al., 2018a;b), Hsu et al. (2019) introduced the notion of a heavy hitter oracle into the data stream model. The heavy hitter oracle, trained on past data, receives a coordinate index $i$ and returns a prediction of whether $x_i$ will be heavy or not. They showed that such an oracle simplifies classical heavy hitter algorithms such as COUNT-MIN and COUNT-SKETCH. We note that similar oracles were also studied in previous work, e.g., membership oracles for Bloom filters by Kraska et al. (2017). These works demonstrate the feasibility of creating such an oracle using machine learning techniques. Recently, learned oracles for low rank approximation, which can be viewed as an analogue of heavy hitters for heavy directions, was explored by Indyk et al. (2019).

The work of Hsu et al. (2019) is the starting point of our work. While Hsu et al. (2019) showed applications of the heavy hitter oracle to frequency estimation, the full power of such an oracle remained largely unexplored. Our main goal is to understand if such an oracle is more broadly useful, and whether it can give memory and time improvements to the large number of data stream problems mentioned above. Accurately estimating the heavy hitters in a vector is the bottleneck of existing algorithms for numerous data stream problems, and one cannot help but ask:

*Can a heavy hitter oracle be applied to a large number of problems in the data stream model to obtain significantly improved bounds for these problems?*

We consider estimating the following common functions $f(x)$ in the data stream model:

- Distinct Elements: $f(x) = \|x\|_0$, where $\|x\|_0$ is the number of nonzero coordinates of $x$, defined as $\|x\|_0 = |\{i : x_i \neq 0\}|$. This quantity is useful for detecing denial of service attacks and database query optimization (Kane et al., 2010b).
- the $F_p$-Moment: $f(x) = \|x\|_p^p$, where $\|x\|_p$ is the usual $\ell_p$-norm of $x$, defined as $\|x\|_p = (\sum_i |x_i|^p)^{1/p}$. For $0 < p < 2$, these are often more robust than the Euclidean norm (Indyk, 2000). For $p > 2$ these are used to estimate skewness and kurtosis (Indyk & Woodruff, 2005).
- the $(k, p)$-Cascaded Norm: in this problem $x$ is an $n \times d$ matrix that receives updates to its individual entries, and $f(x) = \|x\|_{k,p} := (\sum_i (\sum_j |x_{ij}|^p)^{k/p})^{1/k}$. These are useful for executing multiple databse queries (Cormode & Muthukrishnan, 2005; Jayram & Woodruff, 2009).

These problems have also been studied for non-numerical data, such as geometric data. For example, the $F_p$-moment problem was studied on a stream of rectangles in Tirthapura & Woodruff (2012). One can think of an underlying $\Delta^d$-dimensional vector $x$, identified with a $d$-dimensional rectangle $\{1, \ldots, \Delta\}^d$. Without loss of generality, we assume that $\Delta$ is a power of 2. Each item in the stream is a pair of the form $(R, w)$, where $R$ is a $d$-dimensional rectangle and $w$ is an integer, meaning

| Problem | Result Type | Previous Result | This work |
|---|---|---|---|
| Distinct Elements | S | $O(\frac{1}{\epsilon^2}\log n(\log\frac{1}{\epsilon}+\log\log n))$ Kane et al. (2010b) | $O(\frac{1}{\epsilon^2}\log n\log\frac{1}{\epsilon})$ Theorem 3 |
| $F_p$ Moment $(p>2)$ | S | $\widetilde{O}(n^{1-2/p})$ e.g., Andoni et al. (2011) | $\widetilde{O}(n^{1/2-1/p})$ Theorem 7 |
| $F_p$ Moment Fast Update $(p<2)$ | Update T Reporting T | amortized $O(\log^2\frac{1}{\epsilon}\log\log\frac{1}{\epsilon})$ $O(\frac{1}{\epsilon^2}\log^2\frac{1}{\epsilon}\log\log\frac{1}{\epsilon})$ Kane et al. (2011) | expected amortized $O(1)$ $O(\frac{1}{\epsilon^2})$ Theorem 17 |
| Cascaded Norm $(k\geq p\geq 2)$ | S | $\widetilde{O}(n^{1-2/k}d^{1-2/p})$ Jayram & Woodruff (2009) | $\widetilde{O}(n^{1-1/k-p/(2k)}d^{1/2-1/p})$ Corollary 20 |
| Rectangle $F_p$ Moment $(p>2)$ | S | $\widetilde{O}(\Delta^{d(1-2/p)}\operatorname{poly}(\frac{d}{\epsilon}))$ Tirthapura & Woodruff (2012) | $\widetilde{O}(\Delta^{d(1/2-1/p)}\operatorname{poly}(\frac{d}{\epsilon}))$ Theorem 23 |

Table 1: Summary of previous results and the results obtained in this work. We assume that $m, M = \operatorname{poly}(n)$. In the column of result type, S denotes space complexity and T denotes time complexity. We view $\epsilon$ as a constant for the listed results of the $F_p$ Moment and Cascaded Norm problems.

| Problem | Heavy Hitter Oracle |
|---|---|
| Distinct Elements | $|x_i|\geq 2^{\operatorname{poly}(1/\epsilon)}$ |
| $F_p$ Moment $(p>2)$ | $|x_i|^p\geq\frac{1}{\sqrt{n}}\|x\|_p^p$ |
| $F_p$ Moment Fast Update $(p<2)$ | $|x_i|^p\geq\epsilon^2\|x\|_p^p$ $|x_i|^p\geq\frac{\epsilon^2}{\log^2(1/\epsilon)\log\log(1/\epsilon)}\|x\|_p^p$ |
| Cascaded Norm $(k\geq p\geq 2)$ | $|x_i|^p\geq\|x\|_p^p/(d^{\frac{1}{2}}n^{1-\frac{p}{2k}})$ |
| Rectangle $F_p$ Moment $(p>2)$ | $|x_i|^p\geq\|x\|_p^p/\Delta^{d/2}$ |

Table 2: Summary of the threshold of the heavy hitter oracles used in each problem. Note that two oracles are used for the $F_p$ Moment Fast Update problem.

$x_i\leftarrow x_i+w$ for all points $i\in R$. The goal is to estimate $f(x)=\|x\|_p^p$. This is called the Rectangle $F_p$-Moment problem, which occurs in spatial data streams and constraint databases.

**Our Results**  We show that a heavy hitter oracle can greatly improve the complexity of a wide array of commonly studied problems in the data stream model, leading to the first optimal bounds for several important problems, and shattering lower bounds that have stood in the way of making further progress on important problems. We note that not only do we give new algorithms, we also give several lower bounds for algorithms equipped with the oracle, that show optimality of our algorithms even with an oracle. Our algorithms not only give practically better, theoretically-grounded improvements to these problems with an oracle, they also shed light on what exactly the difficulties are of making further progress on data stream problems without an oracle. We consider both perfect oracles and oracles that may sometimes make mistakes.

We summarize in Table 1 our improvements to existing algorithms using appropriate heavy hitter oracles, whose conditions are summarized in Table 2. Throughout, we make the conventional assumption that $m, M = \operatorname{poly}(n)$.

*Distinct Elements:* For the problem of estimating $\|x\|_0$, we improve the prior memory bound of $O(\frac{1}{\epsilon^2}\log n(\log\frac{1}{\epsilon}+\log\log n))$ to $O(\frac{1}{\epsilon^2}\log n\log\frac{1}{\epsilon})$. For constant $\epsilon$, this gives an optimal $O(\log n)$ bits of memory, breaking a recent $\Omega(\log n\log\log n)$ bit lower bound shown in Woodruff & Yang (2019). Examining the lower bound argument in Woodruff & Yang (2019), the hard instance is precisely when there are many items of large frequency, namely, the product of the first small number of primes. This prevents hashing-based algorithms, such as the one in Kane et al. (2010b), from working with $O(\log n)$ bits of space, since hash values are typically taken modulo a small prime and such heavy items necessitate the use of a large enough prime. Surprisingly, we show that with a heavy hitter oracle, we can separately handle such items and thus bypass this lower bound. We note that our $O(\log n)$-bit algorithm is optimal even given a heavy hitter oracle; see the $\Omega(\log n)$ lower bound in Alon et al. (1999), which holds even if all item frequencies are in the set $\{0, 1, 2\}$.

$F_p$-*Moment Estimation,* $0 < p < 2$: There is a long line of work on this problem with the best known bounds given in Kane et al. (2011), which achieve an optimal $O(\epsilon^{-2} \log n)$ bits of space, and $O(\log^2(1/\epsilon) \log \log 1/\epsilon)$ time to process each element. The existing algorithm is based on separately handling those elements $i$ for which $|x_i|^p \geq \epsilon^2 \|x\|_p^p$, i.e., the heavy hitters, from the remaining "light items". Although we can simply store the heavy hitters given a heavy hitter oracle, unfortunately, the large $O(\log^2(1/\epsilon) \log \log 1/\epsilon)$ processing time in the algorithm of Kane et al. (2011) also comes from running a fast multipoint evaluation on the light elements.

We observe that if we instead train *two oracles*, one for detecting heavy hitters, and one for detecting items $x_i$ with $|x_i|^p \geq \frac{\epsilon^2}{\log^2(1/\epsilon) \log \log(1/\epsilon)} \|x\|_p^p$, then we can also separate out "medium items", i.e., items that are detected with this oracle but not our heavy hitter oracle. There are only $O(\epsilon^{-2} \log^2(1/\epsilon) \log \log(1/\epsilon))$ medium items and so assuming $\log n \geq \mathrm{poly}(\log(1/\epsilon))$, one could perfectly hash all such items and maintain their hashed identities in the optimal $O(\epsilon^{-2} \log n)$ bit of space. However, one cannot maintain their counts, since each count is $O(\log n)$ bits, and approximate counts do not work if the stream can increment or decrement the coordinate values $x_i$. This is too much memory, as it would lead to $O(\epsilon^{-2}(\log n) \log^2(1/\epsilon) \log \log(1/\epsilon))$ bits of space, even worse than without the oracle. We could also maintain a so-called $p$-stable random variable for each medium item, and use it to estimate the $F_p$-moment of the medium items, but this would again be $O(\epsilon^{-2}(\log n) \log^2(1/\epsilon) \log \log(1/\epsilon))$ bits of space to store one random variable per medium item. We instead observe that most of the $p$-stable random variables are small, and so can be stored with many fewer than $\log n$ bits, and therefore globally we can get by with only $O(\epsilon^{-2} \log n)$ bits in total for estimating the medium items. Finally, for the light items, the critical observation is that we can sub-sample a $\frac{1}{\log^2(1/\epsilon) \log \log(1/\epsilon)}$ fraction of them, and still accurately estimate their contribution. But this means we can run the multipoint evaluation algorithm of Kane et al. (2011) on such items, since while that has update time $O(\log^2(1/\epsilon) \log \log(1/\epsilon))$, on average it is only executed on one out of every $\log^2(1/\epsilon) \log \log(1/\epsilon)$ items.

In summary, with the aid of our oracles and by separately handling these three types of items, we are able to achieve the optimal $O(1)$ amount of update time per item, while retaining the optimal $O(\epsilon^{-2} \log n)$ bits of memory.

$F_p$-*Moment Estimation for* $p > 2$: It is well-known (Bar-Yossef et al., 2004) that a hard input distribution for this problem is when a random coordinate of $x$ has absolute value $n^{1/p}$, and all other coordinates have value $0$ or $1$. In this case, one needs to certify that there is indeed a coordinate of value $n^{1/p}$, as it contributes a constant fraction of $\|x\|_p^p$. There is a known $\Omega(n^{1-2/p})$ bits of space lower bound for this. However, a heavy hitter oracle exactly allows us to overcome this, as it can be used to separately identify such items. Our idea is to separate out every item $i$ for which $|x_i|^p \geq \frac{1}{n^{1/2}} \|x\|_p^p$ with the oracle, and run an existing $F_p$-estimation algorithm on such items. Since there are at most $n^{1/2}$ items, existing algorithms give $(n^{1/2})^{1-2/p} \mathrm{poly}(\log n) = n^{1/2-1/p} \mathrm{poly}(\log n)$ bits of memory for this problem. For the remaining light elements, one can show that if one subsamples only a $1/n^{1/2}$ fraction of them and then runs an existing $F_p$-estimation algorithm on them, one can use this to accurately estimate their contribution. This again takes $n^{1/2-1/p} \mathrm{poly}(\log n)$ bits of memory, balancing the memory with the other part of our algorithm and giving a quadratic improvement in memory in the case without a heavy hitter oracle. Overall our algorithm uses $O(n^{1/2-1/p} \mathrm{poly}(\log n))$ bits with an oracle and we can in fact show that this is tight up to logarithmic factors as there exists a lower bound of $\Omega(n^{1/2-1/p})$ bits provided the same oracle.

We describe our algorithms for cascaded norms and rectangle $F_p$-moments in the appendix.

We conduct experiments for the distinct elements and the $F_p$ moment ($p > 2$) problems, on both real-world and synthetic data, which demonstrate significant practical benefits.

## 2 PRELIMINARIES

**Notation** We let $[n]$ denote the set $\{1, \ldots, n\}$. The function $\mathrm{lsb}(x)$ denotes the index of the least significant bit of a non-negative integer $x$.

**Space and Time Complexity** For streaming algorithms, the efficiency is characterized by the space (memory) complexity and the time complexity. The time is further divided into update time and reporting time. A data stream algorithm maintains a data structure $\mathcal{D}$ while processing the data stream and at the end of the stream, it outputs an approximation $Z$ to $f(x)$. The space complexity refers to the size, in bits, of $\mathcal{D}$. The update time refers to the time spent to update the data structure $\mathcal{D}$ upon receiving a single stream update. The reporting time refers to the time to output the approximation $Z$ at the end of the stream.

**Heavy Hitter Oracle** We assume access to a heavy hitter oracle, which receives an input $i \in [n]$, and outputs whether $x_i$ will be a heavy hitter at the end of the stream. The definition of heavy hitter varies for different data stream problems. There are two kinds of oracles. The first kind indicates whether or not $|x_i| \geq T$ and the second kind indicates whether or not $|x_i|^p \geq \frac{1}{T}\|x\|_p^p$, where $T$ is a pre-determined threshold associated with the problem. We shall use an oracle of the first kind for the distinct elements problem and oracles of the second kind for all other problems.

In the experiments, the heavy hitter oracle receives an $i \in [n]$ and outputs the predicted value of $x_i$. The implementation of the first kind of oracle is thus straightforward. For the second kind of oracle, there are at most $T$ coordinates $x_i$ satisfying $|x_i|^p \geq \frac{1}{T}\|x\|_p^p$. We shall regard the largest $T$ coordinates in the predicted vector $x$ as the heavy hitters in our implementation (instead of calculating the $\|x\|_p^p$ of the predicted vector $x$), which is in conformity with the implementation in the prior experiments of Hsu et al. (2019).

## 3 DISTINCT ELEMENTS

The current best algorithm for estimating $L_0 = \|x\|_0$, in the presence of additions and deletions to coordinates of $x$, is due to Kane et al. (2010b). A key component of their algorithm is a constant-factor approximation algorithm, called ROUGHL0ESTIMATOR, which returns a value $\widetilde{L}_0$ satisfying $\frac{1}{110}L_0 \leq \widetilde{L}_0 \leq L_0$ with probability at least $2/3$. The space is $O(\log n \log \log(mM))$ bits and the update and reporting times are $O(1)$. In this section, we improve the space of ROUGHL0ESTIMATOR to $O(\log n)$ bits with the help of a trained oracle. All proofs are postponed to Section A.

The algorithm first picks a pairwise hash function $h : [n] \to [n]$ and subsamples $n$ coordinates at $\log n$ scales, where the $j$-th scale consists of the coordinates $i$ for which $\mathrm{lsb}(h(i)) = j$. In each scale, it uses a small-space counting algorithm EXACTCOUNT to obtain, with probability at least $1 - \eta$, the exact value of $\|x\|_0$ at that scale. Then it finds the deepest scale, i.e., the largest $j$, with $\|x\|_0$ above some fixed constant threshold, and returns $\widetilde{L}_0 = 2^j$ as the estimate. If such a $j$ does not exist, it returns $\widetilde{L}_0 = 1$.

The EXACTCOUNT procedure in Kane et al. (2010b) uses a hash function $h' : [n] \to [B]$, where $B = \Theta(c^2)$ for some constant $c$ for which $h'$ is a perfect hashing of a set of size at most $c$. Each bucket $\ell \in [B]$ is a counter which keeps track of $(\sum_{h(i)=\ell} x_i) \bmod p$ for some large prime $p$. The estimate is the number of nonzero buckets, and EXACTCOUNT returns the maximum estimate of $\Theta(\log(1/\eta))$ parallel repetitions.

Now, with the trained oracle, we can pick $p$ to be a smaller $\mathrm{poly}(1/\epsilon)$ value in EXACTCOUNT. In the higher-level ROUGHL0ESTIMATOR algorithm, whenever an update is made to a heavy coordinate $i$ identified by the oracle, the corresponding bucket inside EXACTCOUNT is marked as nonempty regardless of the counter value, since we know the heavy item will never be entirely deleted, since by definition it is heavy at the end of the stream (note this is not true of non-heavy elements). In the end, ROUGHL0ESTIMATOR finds the deepest scale with at least one nonempty bucket.

The following is our new guarantee of EXACTCOUNT.

**Lemma 1.** *Assume that $L_0 \leq c$ and suppose that $B = \Theta(c^2)$. The subroutine EXACTCOUNT returns the exact value of $L_0$ with probability at least $1 - \eta$ using $O(B \log(1/\epsilon))$ bits of space, in addition to storing $O(\log(1/\eta))$ independently chosen pairwise-independent hash functions from $[n]$ to $[B]$. The update and reporting times are $O(1)$.*

Now we specify some details of the algorithm ROUGHTL0ESTIMATOR. It chooses $c = 141$ and $\eta = 1/16$ for EXACTCOUNT, and the EXACTCOUNT algorithms for different scales all share the same $O(\log(1/\eta))$ hash functions. The following is our new guarantee.

**Theorem 2.** *The algorithm* ROUGHL0ESTIMATOR *outputs a value* $\widetilde{L}_0$*, which satisfies* $\frac{1}{110}L_0 \leq \widetilde{L}_0 \leq L_0$ *with probability at least* $13/16$*. The space complexity is* $O(\log(n)\log(1/\epsilon))$ *bits, and the update and reporting times are* $O(1)$*.*

With a constant-factor estimate $R$ to $L_0$, the high-level idea is to subsample each index in $[n]$ with probability $K/R$. Then the number of distinct surviving items is $\Theta(K)$. To estimate the number of distinct surviving items efficiently, one hashes them into $\Theta(K)$ buckets and counts the number $T$ of non-empty buckets, which is simulated by updating a bit vector of length $\Theta(K)$. The final estimate can be deduced from the number of distinct surviving items, which can be shown to be a $(1 + eps)$-approximation to $F_0$ when $K = \Theta(1/\epsilon^2)$. Since we do not know $L_0$ in advance, we have to guess its value using $\log n$ geometrically increasing values, and so our memory can be thought of as a bucket matrix $A$ of dimension $(\log n) \times K$. In parallel we run the ROUGHL0ESTIMATOR above, so at the end of the stream, we will obtain a constant-factor estimate $R$, allowing us to look into the appropriate row of $A$ (the appropriate buckets). A problem is how to efficiently maintain a bucket. In Kane et al. (2010b), each bucket stores the dot product of frequencies hashing to it with a random vector over a large finite field of order $\text{poly}(K\log(nM))$, using $O(\log K + \log\log(nM)) = O(\log(1/\epsilon) + \log\log(nM))$ bits. Now, using a similar argument to the proof of Lemma 1, with the help of our heavy hitter oracle, we can mark the buckets containing a heavy hitter as nonempty and reduce the order of the finite field to $O(\text{poly}(1/\epsilon))$. This improves the space complexity of each bucket to $O(\log(1/\epsilon))$ bits. Combined with the complexity of ROUGHL0ESTIMATOR, we conclude with the following theorem.

**Theorem 3.** *There is an algorithm for* $(1 \pm \epsilon)$*-approximating* $L_0$ *using space* $O(\epsilon^{-2}(\log n)\log(1/\epsilon))$ *with success probability at least* $2/3$*, and with* $O(1)$ *update and reporting times.*

## 4 $F_p$ ESTIMATION, $p > 2$

We start with a well-known algorithm for estimating $\|x\|_p^p$ in a stream.

**Lemma 4** (Precision Sampling, Andoni et al. (2011))**.** *There is a randomized algorithm, called the Precision Sampling algorithm, which returns an estimate $X$ to $\|x\|_p^p$ such that it holds with probability at least* $0.9$ *that* $(1 - \epsilon)\|x\|_p^p \leq X \leq (1 + \epsilon)\|x\|_p^p$*. The algorithm uses $B(n) = O(p^2\epsilon^{-2-4/p}n^{1-2/p}\log(n)\log(M))$ bits of space.*

In the rest of the section, we assume that there is an oracle which can tell us if $|x_i|^p \geq \frac{1}{s}\|x\|_p^p$ is a heavy hitter. The main idea is that we separately estimate the $F_p$-moment of the heavy hitters, and for the remaining light elements, we use sub-sampling to estimate their contribution with sampling rate $1/\rho$.

Let $I_H, I_L \subseteq [n]$ denote the subsets of indices of the heavy hitters and the light elements, respectively. Since $|I_H| \leq s$, we can use an $F_p$ estimator to $(1 + \epsilon)$-approximate $\|x_{I_H}\|_p^p$ with space $B(s)$. For the light elements, we use a sub-sampling algorithm: we set a sampling rate $1/\rho$, and for each item $i \in I_L$, with probability $1/\rho$ we choose it. For each item $i$ we choose, we calculate $F_s = \sum |x_i|^p$, and use $\rho F_s$ to estimation $\|x_{I_L}\|_p^p$. By a Chernoff bound, with high probability, we sample at most $3n/\rho$ elements.

To analyze it, we let $Y = \rho F_S$, and we have the following lemmas. All proofs in this section are postponed to Section B.

**Lemma 5.** $\mathbb{E}[Y] = \|x_{I_L}\|_p^p$, $\mathbf{Var}[Y] \leq \frac{\rho}{s}\|x\|_p^{2p}$.

**Lemma 6.** *If we repeat the sub-sampling $k\rho/s$ times independently with sub-sampling rate $\rho\epsilon^2$, we can get a $(1 \pm \epsilon)$-approximation of $F_p$ with probability at least $1 - 1/k$.*

If we use the estimator in Lemma 4 in both parts (heavy part and light part), The total space complexity is $B(s) + \epsilon^{-2}\frac{\rho}{s}B(\frac{n}{\rho\epsilon^2})$. Letting $\rho = s = \sqrt{n}$, we obtain an $F_p$ estimation algorithm with space $O(\epsilon^{-4}n^{1/2-1/p}\log(n)\log(M))$. We remark that this bound is tight in $n$ up to logarithmic factors as there is a lower bound of $\Omega(n^{1/2-1/p})$ bits even in the presence of the oracle.

**Theorem 7.** *Under the assumption of a heavy hitter oracle, we can estimate $\|x\|_p^p$ within a factor $1 \pm 2\epsilon$ in $O(\epsilon^{-4}n^{1/2-1/p}\log(n)\log(M))$ bits with success probability at least $3/5$.*

The above analysis is based on the assumption that the oracle is perfect. In practice, sometimes the oracle makes mistakes. Here we assume that for each item $i$, the probability the oracle gives us an incorrect prediction is $\delta$. We have the following theorem for noisy oracles.

**Theorem 8.** *With a noisy heavy hitter oracle with error probability $\delta$, we can estimate $\|x\|_p^p$ within a factor $1 \pm 2\epsilon$ in $O(\epsilon^{-4}n^{1/2-1/p}\log(n)\log(M))$ bits of space when $\delta = O(1/\sqrt{n})$, or in $O(\epsilon^{-4}(n\delta)^{1-2/p}\log(n)\log(M))$ bits of space otherwise. Both guarantees hold with success probability at least $3/5$.*

## 5 EXPERIMENTS

### 5.1 DISTINCT ELEMENTS

#### 5.1.1 INTERNET TRAFFIC DATA

For this experiment, the goal is to estimate the number of distinct packets for each network flow. A flow is a sequence of packets between two machines on the Internet. It is identified by the IP addresses of its source and destination and the application ports.

**Dataset**: The traffic data is collected at a backbone link of a Tier1 ISP between Chicago and Seattle in 2016 (CAIDA). Each recording session is around one hour. Within each minute, there are around 30 million packets (meaning 30 million updates) and 1 million unique flows. The distribution of packet counts over Internet flows is heavy tailed, see Hsu et al. (2019).

**Model**: We use the prediction results in Hsu et al. (2019), which predict the logarithm of the packet counts for each flow. In Hsu et al. (2019), the authors use two RNNs to encode the source and destination IP addresses separately and additionally use two-layer fully-connected networks to encode the source and destination ports. Then they concatenate the encoded IP vectors, encoded port vectors, and the protocol type as the final features to predict the packet counts. They use the first 7 minutes for training, the following minute for validation, and estimate the packet counts in subsequent minutes.

**Results**: We plot the estimation error vs. total number of buckets for the 20th minute in the CAIDA data. We run the estimation algorithm 5 times and calculate the average estimation error. The result is shown in Figure 2. The total number of unique flows appearing in a single test minute is about $1.1 \times 10^6$. For every light coordinate, we make an extra update that makes this coordinate 0 with probability 0.5.

There are two parts to our algorithm. For ROUGHL0ESTIMATOR, we set $c = 10$ and $\eta = 1/4$. We use the heavy hitter oracle to predict whether the coordinate will be larger than $2^{10}$. We randomly select a prime from $[11, 31]$ for the hash buckets. The total number of buckets in this part is about $4 \times 10^3$ (each bucket can store a number in $\{0, 1, \dots, 31\}$ and has an extra bit to indicate whether it has been marked). For the other part of our algorithm, we create a bucket matrix $A$ of dimension $(\log n) \times K$ with $K$ ranging from $2^6$ to $2^9$. Each entry in $A$ can also store a number between 0 and 31 and with an additional bit to indicate whether it has been marked. From the plot we can see that the algorithm achieves an approximation error of about 2.5% using at most $1.5 \times 10^4$ buckets, approximately 1.5% of the space needed if we were to keep an individual counter for each flow.

#### 5.1.2 SYNTHETIC DATA

To further demonstrate the advantage of having a heavy hitter oracle, we run the previous algorithm due to Kane et al. (2010b) (where we randomly choose a prime $p \in [11, 31]$ as the divisor for each bucket) and our modified algorithm on the synthetic data designed as follows: first, we generate an input vector $x$ of dimension $n = 10^6$ with i.i.d entries uniform on $\{0, 1, \dots, 100\}$. Then for each coordinate $i$ and each prime $p \in [11, 31]$, we multiply $x_i$ by $p$ with probability 0.2. We set the threshold to be a heavy hitter to be $2^{10}$, that is, the oracle reports $x_i$ as a heavy hitter if $|x_i| > 2^{10}$.

**Results**: We use the optimal parameters in the last experiment ($c = 10$, $\eta = 1/4$ and $K = 2^9$) and vary the total number of buckets from 7000 to 15000. The results are shown in Figure 1. We see that having an oracle significantly reduces the estimation error.

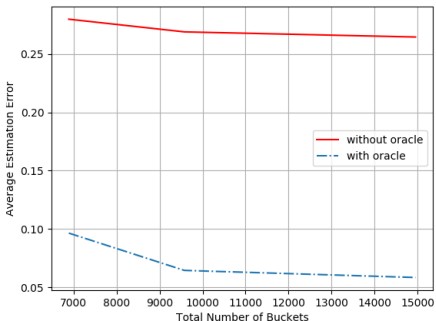

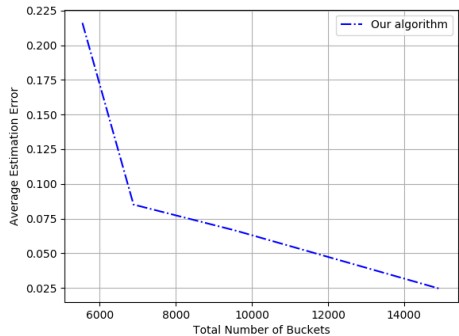

Figure 1: The estimation error of the $L_0$-norm in the synthetic data

Figure 2: The estimation error of the $L_0$-norm in the 20th minute of the CAIDA data

## 5.2 $F_p$ ESTIMATION

### 5.2.1 SEARCH QUERY DATA

For this experiment, the goal is to estimate the frequency moments of the vector indicating the number of occurrences each search query appears.

**Dataset**: We use the AOL query log dataset, which consists of 21 million search queries collected from 650 thousand users over 90 days. There are 3.8 million unique queries. Each query is a search phrase with multiple words. The distribution of the search query frequencies is Zipfian, see Hsu et al. (2019).

**Model**: We use the prediction result in Hsu et al. (2019), which predicts the number of times each search phrase appears. In Hsu et al. (2019), the authors train an RNN with LSTM cells that takes characters of a search phrase as input. The final states encoded by the RNN are fed to a fully-connected layer to predict the query frequency. They use the first 5 days for training, the following day for validation, and estimate the number of times different search queries appear in subsequent days. We use this prediction result as an indicator of which search phrases are the heavy hitters.

**Results**: We plot the estimation error versus the total number of buckets for the 50th and 80th day ($p = 3$ or $p = 4$). For the same data, we run the estimator 10 times and calculate the average estimation error. The results are shown in Figure 3.

For our algorithm, we notice that the total number of search phrases that appear in a single day is about $n = 1.5 \times 10^5$ and so $\sqrt{n} < 400$. We can store the heavy coordinates identified by the heavy hitter oracle directly. Given the budget of the number of buckets, we divide them into two parts. Half of them were used to store the heavy items, and the other half are used by the sub-sampling algorithm with the precision sampling estimators to estimate the frequency moment of the light elements.

In comparison, we also run the classical precision sampling estimators. Our results show a larger estimation error owing to the constraint of relatively few buckets. Note that for a fixed bucket budget $B_{\text{tot}}$, there is a trade-off between the number $k$ of repetitions and the number $B = B_{\text{tot}}/k$ of buckets allotted for the hash function in each repetition. We plot the results for different values of $k = 10, 20, 30$.

It is clear from the plots that the estimation error of the oracle-aided algorithm is about 1% even when the total number of buckets is small, demonstrating a strong advantage over classical precision sampling estimators.

### 5.2.2 SYNTHETIC DATA

A number of real-world datasets have the property that the heavy coordinates contribute a large fraction of the $F_p$-moment, as $p$ increases, and so one needs only to obtain a good approximation of

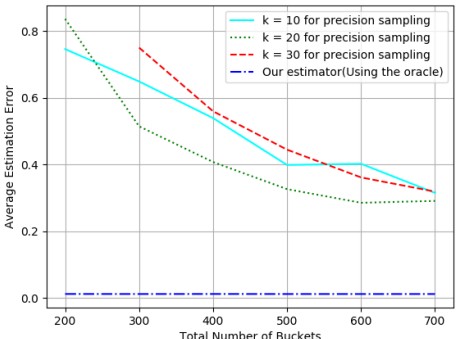 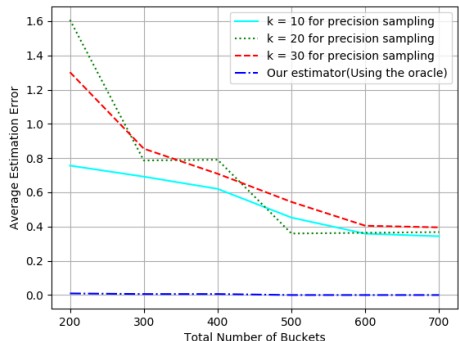

Figure 3: The estimation error of the moment of search queries. The left figure corresponds to $p = 3$ (80th test day) and the right $p = 4$ (50th test day).

the $F_p$-moment of the heavy coordinates. This is indeed one of the reasons why we achieved low error for search query estimation. In order to show that our subsampling algorithm also achieves a good approximation for the light coordinates, we considered the following data set.

We generated an input vector $x$ of dimension $n = 10^8$ with the first $\sqrt{n}$ coordinates being heavy and the remaining coordinates being light. Each coordinate is a uniform random variable in $(0, 1)$, later renormalized such that the heavy and the light parts contribute equally to the total $F_p$-moment, i.e., $\|x_{I_H}\|_p^p = \|x_{I_L}\|_p^p$. Hence, a good approximation of $\|x\|_p^p$ requires a good approximation of both parts.

**Results**: We plot the estimation error vs. total number of buckets for $p = 3$ in Figure 4. For our algorithm and classical precision sampling, we fixed the number $k$ of repetitions to 100 and varied the total number of buckets from $10^6$ to $2 \times 10^6$ (1% to 2% of the length of $x$). We repeated the sub-sampling algorithm 9 times. In each of the 9 repetitions, we also used the precision sampling estimators, repeating 100 times.

From the plots we see that our algorithm achieves significantly smaller relative approximation error than the classical precision sampling algorithm for every number of buckets. In particular, using $2 \cdot 10^6$ buckets, i.e., 2% space of what is needed to store the entire vector $x$, our algorithm achieves a 15% relative estimation error while the classical precision sampling obtains only a 27% relative error. Another observation is that since subsampling reduces the space consumption, our algorithm has a faster update time when $n$ is large.

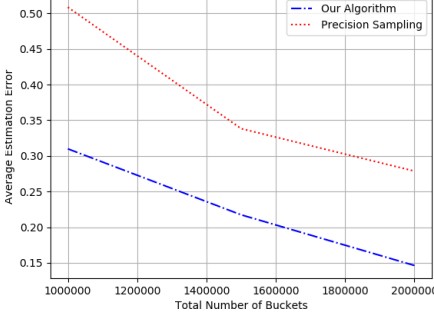

Figure 4: Estimation error for the $F_3$-moment on synthetic data

ACKNOWLEDGMENTS

The authors would like to thank the anonymous reviewers for helpful comments. Y. Li was supported in part by Singapore Ministry of Education (AcRF) Tier 2 grant MOE2018-T2-1-013. D. Woodruff would like to thank partial support from the National Science Foundation under Grant No. CCF-1815840 and the Office of Naval Research (ONR) under grant N00014-18-1-2562.

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

## A  OMITTED PROOFS IN SECTION 3

*Proof of Lemma 1.* If a bucket contains a heavy hitter, it is marked nonempty and it is indeed nonempty. It suffices to show that a bucket containing only light coordinates is nonempty modulo $p$. Since the light coordinates are at most $2^{\mathrm{poly}(1/\epsilon)}$, there are at most $c$ of them by assumption, their sum $S$ is thus at most $2^{\mathrm{poly}(1/\epsilon)}$. This implies that $S$ has at most $\mathrm{poly}(1/\epsilon)$ prime factors. If we choose a random prime $p$ of size $\mathrm{poly}(1/\epsilon)$, then $S \bmod p \neq 0$ with probability at least $1 - \mathrm{poly}(\epsilon)$ if $S \neq 0$. Taking a union bound over $\Theta(c^2)$ buckets proves the correctness. The space and time complexity are immediate from the description of the algorithm. $\square$

*Proof of Theorem 2.* The proof is almost identical to that in Kane et al. (2010b). The space and time complexities are follow from the description of the algorithm.

Next we show correctness. Let $L_0(j)$ denote the true number of distinct elements in the $j$-th scale. Then $\mathbb{E}\, L_0(j) = L_0/2^j$. Let $j^* = \max\{j : \mathbb{E}\, L_0(j) \geq 1\}$ and $j^{**} = \max\{j < j^* : \mathbb{E}\, L_0(j) \geq 55\}$. As shown in Kane et al. (2010b), if $j^{**}$ exists, it holds that $55 \leq \mathbb{E}\, L_0(j^{**}) < 110$ and $\Pr\{32 < L_0(j^{**}) < 142\} \geq 8/9$. With our choices of $c = 141$ and $\eta = 1/16$, EXACTCOUNT returns a nonzero value for the $j^{**}$-th scale with probability $8/9 - 1/16 > 13/16$ by Lemma 1. It follows that the deepest scale $j$ we find satisfies $j^{**} \leq j \leq j^*$. Hence, $L_0 = 2^{j^{**}} \mathbb{E}\, L_0(j^{**}) \leq 110 \cdot 2^j$ and $L_0 = 2^{j^*} \mathbb{E}\, L_0(j^*) \geq 2^j$ as desired. If $j^{**}$ does not exist, then $L_0 < 55$, and $\widetilde{L}_0 = 1$ is a 55-approximation in this case. $\square$

## B  OMITTED PROOFS IN SECTION 4

*Proof of Lemma 5.* We let $Y = \rho F_s = \rho \sum_i f_i |x_i|^p$, where $f_i = 1$ if $i$ was chosen, and $f_i = 0$ otherwise. So we have that $\mathbb{E}[f_i] = \frac{1}{\rho}$.

It thus follows that $\mathbb{E}[Y] = \|x_{I_L}\|_p^p$, that is, $Y$ is an unbiased estimate. Then,

$$\mathbf{Var}[Y] = \mathbb{E}[Y^2] - \mathbb{E}[Y]^2 = \rho^2 \left( \mathbb{E}[\sum_i f_i |x_i|^{2p} + \sum_{i \neq j} f_i f_j |x_i|^p |x_j|^p] \right) - \|x\|_p^{2p}$$

$$\leq \rho^2 \left( \mathbb{E}[\sum_i f_i |x_i|^{2p}] + \sum_{i,j} \frac{1}{\rho^2} |x_i|^p |x_j|^p \right) - \|x\|_p^{2p}$$

$$= \rho \sum_{i \in I_L} |x_i|^{2p} .$$

To bound this, we notice that we have the fact that $x_i < \frac{\|x\|_p^p}{s}$ and $p > 2$, so

$$\rho \sum_{i \in I_L} x^{2p} \leq \rho s \frac{\|x\|_p^{2p}}{s^2} = \frac{\rho}{s} \|x\|_p^{2p}.$$

So finally we have that $\mathbf{Var}[Y] \leq \frac{\rho}{s} \|x\|_p^{2p}$ . $\square$

*Proof of Lemma 6.* By Chebyshev's inequality, we have that

$$\Pr\left\{ |\overline{Y} - \|x\|_p^p| > \epsilon \|x\|_p^p \right\} \leq \frac{\frac{\rho \epsilon^2}{ks} \|x\|_p^{2p}}{\frac{\rho}{s} \epsilon^2 \|x\|_p^{2p}} = \frac{1}{k}. \qquad \square$$

*Proof of Theorem 7.* For the heavy hitters, note that $|I_H| \leq n/s = \sqrt{n}$. For this part, we can use $9B(\sqrt{n})$ space to get an $H$ for which $|H - \|x_{I_H}\|_p^p| \leq \epsilon \|x_{I_H}\|_p^p$ with probability at least $\frac{7}{8}$ by independently running 9 estimators and then taking the median.

For the light elements, for every sub-sample round $i$, we independently run $O(1)$ estimators with rate $\sqrt{n}\epsilon^2$ and use their median $Z_i$ to be the estimate of this level of sub-sampling. Let $Y_i$ be the

true $F_p$-moment of this sub-sample. Then with probability at least $\frac{99}{100}$ it holds that $|Y_i - Z_i| \leq \epsilon Y_i$. Taking a union bound, we get that if we do the sub-sampling 10 times, with probability at least $\frac{9}{10}$, $|Y_i - Z_i| \leq \epsilon \|x_{I_L}\|_p^p$ holds for every $i$. Using Lemma 6 we can get that with probability at least $\frac{9}{10}$, $|\overline{Z} - \|x_{I_L}\|_p^p| \leq \epsilon \|x_{I_L}\|_p^p + \epsilon \|x\|_p^p$. The overall success probability of this part is at least $\frac{4}{5}$.

So we get that $|H + Z - \|x\|_p^p| \leq \epsilon \|x_{I_L}\|_p^p + \epsilon \|x\|_p^p + \epsilon \|x_{I_H}\|_p^p = 2\epsilon \|x\|_p^p$ with success probability at least $3/5$. The total space complexity is $B(\sqrt{n}) + O(\epsilon^{-2})B(\frac{\sqrt{n}}{\epsilon^2}) = O(\epsilon^{-4} n^{\frac{1}{2} - \frac{1}{p}} \log(n) \log(M))$.
□

*Proof of Theorem 8.* For an element $x_i$ which the oracle indicates as light, we know that with probability $1 - \delta$, it is truly light. Combined with Theorem 5, we obtain that

$$\mathbf{Var}[Y] \leq \rho(1 - \delta) \sum_{i \in I_L} |x_i|^{2p} + \rho \delta \sum_{i \in I_H} |x_i|^{2p} \leq (1 - \delta)\frac{\rho}{s} \|x\|_p^{2p} + \rho\delta \|x\|_p^{2p}.$$

Hence, when $\delta = O(\frac{1}{\sqrt{n}})$, Theorem 7 continues to hold. Otherwise we can let $s = n\delta$ and $\rho = \epsilon^2 \delta^{-1}$, and the space complexity becomes $O(\epsilon^{-4}(n\delta)^{1 - \frac{2}{p}} \log(n) \log(M))$ bits. □

**Lower Bound** The upper bound in Theorem 7 is asymptotically tight in $n$ up to logarithmic factors, since we have the following matching lower bound even in the presence of an oracle which indicates whether $|x_i| > \frac{1}{\sqrt{n}} \|x\|_p^p$.

**Theorem 9.** *Suppose that $1/\sqrt{n} < \epsilon < 1/4$ and that a heavy hitter oracle indicates whether $|x_i| > \frac{1}{\sqrt{n}} \|x\|_p^p$ for the whole underlying vector $x$. Any randomized streaming algorithm that estimates $\|x\|_p^p$ with a factor $(1 \pm \epsilon)$ with probability $\geq 9/10$ requires $\Omega(\epsilon^{-2/p} n^{1/2 - 1/p})$ bits of space.*

*Proof of Theorem 9.* We assume $\sqrt{n}$ to be an integer and define our hard instance as follows: for every $1 \leq i \leq \sqrt{n}$, we let $x_i = 1$ with probability $1 - 1/\sqrt{n}$, and $x_i = (4\epsilon)^{1/p} n^{1/2p}$ with probability $1/\sqrt{n}$. And for $i > \sqrt{n}$, $x_i = x_{i \bmod \sqrt{n}}$. In this case, $\|x\|_p^p \geq n$, so any coordinate will not be a heavy hitter. We note that if our algorithm can outputs an $(1 \pm \epsilon)$ approximation of $\|x\|_p^p$, it can also output an $(1 \pm \epsilon)$ approximation of $\|x_{[\sqrt{n}]}\|$.

To prove our lower bound, we will use the $\ell_\infty^k$ communication problem in Bar-Yossef et al. (2004): there are two parties, Alice and Bob, holding vectors $a, b \in \mathbb{Z}^m$ respectively, and their goal is to decide if $\|a - b\|_\infty \leq 1$ or $\|a - b\|_\infty \geq k$. From Bar-Yossef et al. (2004), we can get an $\Omega(m/k^2)$ bits lower bound.

Now we claim that: if we set $m = \sqrt{n}$, $k = (4\epsilon)^{1/p} n^{1/2p}$ and make $a - b = x_{[\sqrt{n}]}$, then any streaming algorithm outputs a $(1 \pm \epsilon)$ approximation $Y$ of $\|x_{[\sqrt{n}]}\|_p^p$ can be used to build a communication protocol for solving this $\ell_\infty^k$ problem with communication proportional to the algorithm's space complexity. In this case, if $\|a - b\|_\infty = 1$, then we have $Y \leq (1 + \epsilon)\|x_{[\sqrt{n}]}\|_p^p \leq (1 + \epsilon)\sqrt{n}$. Otherwise, $Y \geq (1 - \epsilon)(\sqrt{n} - 1 + 4\epsilon\sqrt{n}) > (1 + \epsilon)\sqrt{n}$.

By this reduction, we can finally get an $\Omega(m/k^2) = \Omega(\epsilon^{-2/p} n^{1/2 - 1/p})$ bits space lower bound under the oracle assumption. □

## C  FAST MOMENT ESTIMATION, $0 < p < 2$

When $0 < p < 2$, the $F_p$ estimation problem can be done in an optimal $O(\epsilon^{-2} \log n)$ bits of space. Here we show how to improve the update time while maintaining this amount of space.

In the rest of the section, we assume there are two oracles: one can tell us whether $|x_i|^p \geq \epsilon^2 \|x\|_p^p$, and the other can tell us whether $|x_i|^p \geq \frac{\epsilon^2}{\log^2(1/\epsilon) \log\log(1/\epsilon)} \|x\|_p^p$. We call these 3 categories of elements heavy elements, medium elements, and light elements. Let $I_H, I_M, I_L \subseteq [n]$ denote the subset of indices of the heavy elements, medium elements, and the light elements, respectively.

For the heavy elements, we can use $O(\epsilon^{-2} \log(M))$ space to store them. We now turn to the other two parts. For the light elements, we will use the LightEstimator in Section 3 in Kane et al. (2011):

**Lemma 10** (Kane et al. (2011))**.** *Suppose we are given $0 < \epsilon < 1$, and given a list $L \subseteq [n]$ such that $i \in L$ if and only if $|x_i|^p \geq \epsilon^2 \|x\|_p^p$. There is an algorithm* LightEstimator *which returns an estimate $X$ to $\|x_{n \setminus L}\|_p^p$ such that $|X - \|x_{n \setminus L}\|_p^p| \leq \epsilon \|x\|_p^p$ with probability at least $0.9$. The space usage is $O(\epsilon^{-2} \log(nmM))$, the amortized update time is $O(\log^2(1/\epsilon) \log \log(1/\epsilon))$, and the reporting time is $O(1/\epsilon^2)$.*

For the light elements $x_i$, note we have the condition $|x_i|^p \leq \frac{\epsilon^2}{\log^2(1/\epsilon) \log \log(1/\epsilon)} \|x\|_p^p$. Using Lemma 5 Lemma 6 and the argument in Theorem 7, if we use the LightEstimator to do the sub-sampling algorithm with rate $\rho = \log^2(1/\epsilon) \log \log(1/\epsilon)$ for the light elements, we just need to sub-sample $O(1)$ times to get an approximation $X$ of $\|x_{I_L}\|_p^p$ such that $|X - \|x_{I_L}\|_p^p| \leq \epsilon \|x\|_p^p$ with probability at least $0.9$. Note that for each update $(i, v)$ in the sub-sampling process, if $x_i$ is not selected by the sub-sampling algorithm, the update time is $O(1)$ (we do not need to do anything), otherwise, by Kane et al. (2011) we know that the amortized update time is $O(\log^2(1/\epsilon) \log \log(1/\epsilon))$. Notice that in Lemma 5 we just need a pairwise independent hash function for the sub-sampling. So during the process, the expected amortized update time is $O(\frac{\log^2(1/\epsilon) \log \log(1/\epsilon)}{\rho}) = O(1)$.

**Theorem 11.** *Suppose we are given $0 < \epsilon < 1$, and given the lists $I_L$. There is an algorithm which returns an estimate $X$ to $\|x_{I_L}\|_p^p$ such that $|X - \|x_{I_L}\|_p^p| \leq \epsilon \|x\|_p^p$ with probability at least $0.9$. The space usage is $O(\epsilon^{-2} \log(nmM))$, the expected amortized update time is $O(1)$, and the reporting time is $O(1/\epsilon^2)$.*

It remains to handle the medium elements. Here we will still use the LightEstimator, but do some adjustments. Before describing our LightEstimator data structure, we first define the *p-stable distribution*.

**Definition 1** (Zolotarev Zolotarev (1986))**.** *For $0 < p < 2$, there exists a probability distribution $\mathcal{D}_p$ called the p-stable distribution satisfying the following property. For any positive integer $n$ and vector $x \in \mathbb{R}_n$, if $Z_1, ..., Z_n \sim \mathcal{D}_p$ are independent, then $\sum_{j=1}^n Z_j x_j \sim \|x\|_p Z$ for $Z \sim \mathcal{D}_p$.*

**Lemma 12** (Pagh and Pagh, Ostlin & Pagh (2003) Theorem 1.1)**.** *Let $S \subseteq U = [u]$ be a set of $z > 1$ elements, and let $V = [v]$, with $1 < v \leq u$. Suppose the machine word size is $\Omega(\log(u))$. For any constant $c > 0$ there is a word RAM algorithm that, using time $\log(z) \log^{O(1)}(v)$ and $O(\log(z) + \log \log(u))$ bits of space, selects a family $\mathcal{H}$ of functions from $U$ to $V$ (independent of $S$) such that:*

1. *With probability $1 - O(1/z^c)$, $\mathcal{H}$ is z-wise independent when restricted to $S$.*

2. *Any $h \in \mathcal{H}$ can be represented by a RAM data structure using $O(z \log(v))$ bits of space, and $h$ can be evaluated in constant time after an initialization step taking $O(z)$ time.*

In Kane et al. (2011), LightEstimator was defined by creating $R = 4/\epsilon^2$ independent instantiations of the estimator $D_1$, which we label $D_1^1, ..., D_1^R$, and picking a hash function $h : [n] \rightarrow [R]$ using Lemma 12. Upon receiving an update to $x_i$ in the stream, the update was fed to $D_1^{h(i)}$. The estimator $D_1^i$ is a slightly modified version of the geometric mean estimator of Li (2008), which takes a matrix $A \in \mathbb{R}^{t \times n}$, where each row contains $\Omega(1/\epsilon^p)$-wise p-stable entries and the rows being independent from each other, and maintains $y = Ax$ in the stream. Furthermore, in parallel we run the algorithm of Kane et al. (2010a) with constant error parameter to obtain a value $\widetilde{F}_p$ in $[\|x\|_p^p/2, 3\|x\|_p^p/2]$. The estimator $\mathsf{Est}_p$ is $\min\{C_{t,p}(\prod_{j=1}^t |y_j|^{p/t}), \widetilde{F}_p/\epsilon\}$. Hence in Kane et al. (2011), the update time is the time to evaluate a $\Theta(1/\epsilon^p)$-wise independent hash function over a field of size $poly(nmM)$, which is $O(\log^2(1/\epsilon) \log \log(1/\epsilon))$. Therefore, if we can store the hash values of the p-stable entries for the medium elements, then we can improve the update time to $O(1)$ for this part (note that the parameter $t$ is a constant). We need the following lemma.

**Lemma 13.** *Let $k = O(\frac{1}{\epsilon^2} \log^2 \frac{1}{\epsilon} \log \log \frac{1}{\epsilon})$ and $X_1, X_2, ..., X_k$ be p-stable random variables. With probability at least $0.9$, the following holds. There exists $T \subseteq [k]$ such that $|T| = O(1/\epsilon^2)$ for which $|X_i| \leq \text{poly} \log(1/\epsilon)$ for all $i \notin T$, and consequently, each such $X_i$ can be approximated by some $\widetilde{X}_i$ satisfying $|X_i - \widetilde{X}_i| \leq \text{poly}(\epsilon)$ and each such $\widetilde{X}_i$ can be recovered from $O(\log(1/\epsilon))$ bits.*

To prove this, we need the following result.

**Lemma 14** (Nolan (2003), Theorem 1.12). *For fixed $0 < p < 2$, the probability density function $\phi_p$ of the $p$-stable distribution satisfies $\phi_p(x) = O(1/(1 + |x|^{p+1}))$ and is an even function. The cumulative distribution function satisfies $\Phi_p(x) = O(|x|^{-p})$.*

*Proof of Lemma 13.* Let $I = [-\log^{3/p}(1/\epsilon), \log^{3/p}(1/\epsilon)]$. For a $p$-stable random variable $X_i$, by Lemma 14, $\Pr\{X_i \notin I\} = O(1/\log^3(1/\epsilon))$. So the expected number of $X_i$ for which $X_i \notin I$ is $O(k/\log^3(1/\epsilon)) = O(1/\epsilon^2)$. By Markov's inequality, with probability at least $0.9$, the number of these $X_i$ are $O(1/\epsilon^2)$. For $X_i$ contained in $I$, we can uniformly partition $I$ into subintervals of length $\mathrm{poly}(\epsilon)$, and there will be $O(\mathrm{poly}(1/\epsilon))$ many subintervals. Let $\widetilde{X}_i$ be the nearest partition point. It is clear that $X_i$ can be recovered from $O(\log(1/\epsilon))$ bits to store the index of the partition point. $\square$

The following lemma tells us that we can use the approximation value given by the $p$-stable random variables in the $\mathsf{Est}_i$.

**Lemma 15.** *Let $k = O(\frac{1}{\epsilon^2} \log^2 \frac{1}{\epsilon} \log \log \frac{1}{\epsilon})$. Suppose we are given $a, b, x \in \mathbb{R}^k$ and $|a_i - b_i| \le \epsilon^q$ for all $i$. Then we have $|\langle a, x\rangle - \langle b, x\rangle| \le \epsilon^{q-2}\|x\|_p$ for sufficiently small $\epsilon$.*

*Proof.* By the Cauchy-Schwarz inequality, $|\langle a - b, x\rangle| \le \|a - b\|_2 \|x\|_2 \le \sqrt{k\epsilon^{2q}}\|x\|_2 \le \epsilon^{q-2}\|x\|_2 \le \epsilon^{q-2}\|x\|_p$ (recall that $p < 2$). $\square$

Finally, we have the following theorem.

**Theorem 16.** *Suppose we are given $0 < \epsilon < 1$ and the lists $I_M$. There is an algorithm which returns an estimate $X$ to $\|x_{I_M}\|_p^p$ such that $|X - \|x_{I_M}\|_p^p| \le 2\epsilon\|x\|_p^p$ with probability at least $0.8$. The space usage is $O(\epsilon^{-2}\max\{\log(nmM), \log^3(1/\epsilon)\log\log(1/\epsilon)\})$, the amortized update time is $O(1)$, and the reporting time is $O(1/\epsilon^2)$.*

*Proof.* Let $k = \frac{1}{\epsilon^2}\log^2\frac{1}{\epsilon}\log\log\frac{1}{\epsilon}$. Note that there are at most $k$ medium elements. Using Lemma 12, we pick two hash functions $h_1 : [n] \to [k]$, $h_2 : [n] \to [1/\epsilon^2]$, and we can assume with high probability, they perfectly hash our input set of items. Lemma 13 tells us with probability at least $0.9$, we can use these two hash functions and use $O(1)$ time to read the $p$-stable random variables for the medium elements (if $X_i \in I$, we store $X_i$ in the cell which $h_1(i)$ refers to, using $O(\log(1/\epsilon))$ bits. Otherwise we store $X_i$ in the cell which $h_2(i)$ refers to, using $O(\log(M))$ bits).

Suppose that $\phi(x)$ is the pdf of the $p$-stable distribution. For a $p$-stable $Y$, we have that $\Pr\{|Y| < \epsilon^q\} = \int_{-\epsilon^q}^{\epsilon^q}\phi(x)dx \le 2\phi(0)\epsilon^q = \mathrm{poly}(\epsilon)$ when $q$ is sufficiently large. Hence, in Lemma 13 and Lemma 15, we can pick a sufficiently large $q$ such that with probability at least $1 - \mathrm{poly}(\epsilon)$, all of the true values $y_i$ output by the $\mathsf{Est}_i$ are at least $\epsilon^q\|x\|_p$, and the $\widetilde{y_i}$ output by the $\mathsf{Est}_i$ using the stored $p$-stable random variables satisfy $|y_i - \widetilde{y_i}| \le \epsilon^{q+\frac{1}{p}}\|x\|_p$. So the approximation from the $p$-stable random variables will affect our estimate by at most $\epsilon\|x\|_p^p$. This completes the proof. $\square$

Using Theorem 11 and 16 we will have our result:

**Theorem 17.** *Let $0 < p < 2$ and $0 < \epsilon < 1/2$. There exists a randomized algorithm which outputs $(1 \pm 3\epsilon)\|x\|_p^p$ with probability at least $0.7$ using $O(\epsilon^{-2}\max\{\log(nmM), \log^3(1/\epsilon)\log\log(1/\epsilon)\})$ bits of space. The expected amortized update time is $O(1)$. The reporting time is $O(1/\epsilon^2)$.*

In the case of a noisy oracle, we assume that for a fixed $s$, our oracle will not identify more than $O(s)$ items $x_i$ as heavy hitters (meaning that $|x_i| > \frac{1}{s}\|x\|_p^p$). Then we can adapt the preceding theorem to the noisy oracle version.

**Theorem 18.** *Let $0 < p < 2$ and $0 < \epsilon < 1/2$. Suppose that the heavy hitter oracle errs with probability $\delta = O(\frac{\epsilon^2}{\log^2(1/\epsilon)\log\log(1/\epsilon)})$. There exists an algorithm which outputs $(1 \pm 3\epsilon)\|x\|_p^p$ with probability at least $0.7$ using $O(\epsilon^{-2}\max\{\log(nmM), \log^3(1/\epsilon)\log\log(1/\epsilon)\})$ bits of space. The expected amortized update time is $O(1)$. The reporting time is $O(1/\epsilon^2)$.*

*Proof.* We only need to consider the difference in the estimation of $\|x_{I_L}\|$. When $\delta = O(\frac{\epsilon^2}{\log^2(1/\epsilon)\log\log(1/\epsilon)})$, note that our sub-sampling rate is $\log^2(1/\epsilon)\log\log(1/\epsilon)$. It follows from Theorem 8 that Theorem 17 continues to hold. $\square$

# D    CASCADED NORMS

For notational simplicity, we consider approximating $F_k(F_p(x)) = \sum_{i=1}^n (\sum_{j=1}^d |x_{ij}|^p)^k$ with $k \geq 1$ and $p \geq 2$. The cascaded norm $\|x\|_{k,p}$ then satisfies $\|x\|_{k,p}^k = F_{k/p}(F_p(x)$ for $k \geq p \geq 2$.

We follow the algorithm in Jayram & Woodruff (2009), which first downsamples the rows in $\log n$ levels, and then samples in each level $Q = O(n^{1-1/k})$ entries from the row-sampled submatrix proportional to $|x_i|^p$, and at last estimates the $F_k(F_p)$ cascaded norm from those samples. The first step of $\ell_p$-sampling is further decomposed into two steps: (i) dividing the entries into layers by their magnitudes and uniformly sampling a sufficiently large set of entries in each layer, and (ii) performing "$\ell_p$-sampling" from those samples. We shall improve the space complexity of Step (i) with a heavy hitter oracle.

We first elaborate how Step (i) was done. It splits the entries in the row-sampled matrix $X$ into layers, where the $t$-th layer $S_t(X)$ consists of entries in $[\zeta\eta^{t-1}, \zeta\eta^t]$. A layer $t$ is said to be contributing if $|S_t(X)|(\zeta\eta^t)^p \geq F_p(X)/(B\theta)$. It needs to return $\beta_t = \theta Q|S_t(X)|(\zeta\eta^t)^p/F_p(X)$ samples from each contributing layer $t$. Suppose that $|S_t(X)|/2^j \leq \beta_t < |S_t(X)|/2^{j-1}$ for some $j$, and the algorithm subsamples each entry with probability $1/2^j$, obtaining a subset $Y_j$ of entries. It can be shown that with high probability, for a contributing layer $t$, it holds that $(\zeta\eta^t)^2 \geq \|Y_j\|_2^2/(Q^{2/p}\theta^4|X|^{1-2/p}\log|X|)$, and therefore the standard COUNT-SKETCH algorithm can recover all those entries with $\widetilde{O}(Q^{2/p}|X|^{1-2/p})$ space.

Consider the top level without subsampling of the rows. Suppose that the heavy hitter oracle determines whether $|x_i|^p \geq \|x\|_p^p/T$ for each entry $x_i$ of the matrix $x$ with some threshold $T$. In the algorithm of Step (i), without loss of generality, we can assume each layer contains either heavy entries only or light entries only, and we call the layer a heavy layer or a light layer accordingly. If a light layer is contributing, since each entry is at most $F_p/T$, there must exist at least $N_t = T|S_t(X)|(\zeta\eta^t)^p/F_p$ entries in that level. We would require that $N_t \geq \beta_t$, for which it suffices to have $T = \Omega(\theta Q)$. We can downsample the layer at rate $\theta Q/T$, and then apply the preceding algorithm and thus the number of buckets can be reduced by a factor of $T/(\theta Q)$. Note that the heavy layers come from the at most $T$ heavy hitters. This leads to a space complexity of $\widetilde{O}(Q^{2/p}T^{1-2/p} + Q^{2/p}|X|^{1-2/p}/(T/(\theta Q)))$. Let $T = \Theta((|X|Q)^{1/2})$. In this case, $T = \Omega(\theta Q)$ is satisfied and the space complexity is $\widetilde{O}(Q^{1/2+1/p}|X|^{1/2-1/p})$.

For the $j$-th level of row sampling, note that the same (global) heavy hitter oracle, when applied to this row-subsampled matrix, effectively has $T$ replaced with $T/2^j$. However, $|X|$ is also replaced with $|X|/2^j$, and thus we would require $T = \Omega(2^j\theta Q)$ and we can downsample the layer at rate $2^j\theta Q/T$. This means that the space complexity is a $(2^{-j})^{1-2/p}$ fraction of the space of the top level. The constraint of $T = \Omega(2^j\theta Q)$ allows for $j \leq \frac{1}{2}\log\frac{|X|}{Q}$. For $j > \frac{1}{2}\log\frac{|X|}{Q}$, we can just run the old algorithm with space $\widetilde{O}(Q^{2/p}(|X|/2^j)^{1-2/p}) = \widetilde{O}(Q^{1/2+1/p}|X|^{1/2-1/p})$.

Plugging in $Q = \Theta(n^{1-1/k})$ and $|X| = nd$ and noting that the space of the subsequent steps are dominated by this very first sampling step, we have:

**Theorem 19.** *Let $\epsilon > 0$ and $k \geq 1, p \geq 2$ be constants. There exists a randomized algorithm which receives an $n \times d$ matrix $x$ and outputs a $(1+\epsilon)$-approximation to $F_k(F_p(x))$ with probability at least $2/3$ using $\widetilde{O}(n^{1-\frac{1}{kp}-\frac{1}{2k}}d^{\frac{1}{2}-\frac{1}{p}})$ space.*

Replacing $k$ with $k/p$, we obtain our final result for cascaded norms.

**Corollary 20.** *Let $\epsilon > 0$ and $k \geq p \geq 2$ be constants. There exists a randomized algorithm which receives an $n \times d$ matrix $x$ in a stream of additions and deletions to its coordinates, and outputs a $(1+\epsilon)$-approximation to $\|x\|_{k,p}$ with probability at least $2/3$ using $\widetilde{O}(n^{1-\frac{1}{k}-\frac{p}{2k}}d^{\frac{1}{2}-\frac{1}{p}})$ space.*

# E  RECTANGULAR $F_p$

The rectangle-efficient algorithm in Tirthapura & Woodruff (2012) is based on the $F_p$ moment estimation algorithm in Braverman & Ostrovsky (2010). In Braverman & Ostrovsky (2010), they first subsample to obtain the substream $D_j$ ($j = 0, 1, 2, ... \log n$), where $D_0$ is the full stream, and if $i \in D_j$, then $i \in D_{j+1}$ with probability $1/2$. For every substream $D_j$, they run the COUNT-SKETCH algorithm to recover all $(\alpha^{\frac{2}{p}}/n^{1-\frac{2}{p}})$-heavy hitters, whence they can obtain a good final estimate.

The algorithm in Tirthapura & Woodruff (2012) is similar. Instead of updating the counter in each coordinate inside a rectangle, they developed a rectangle-efficient data structure called RECT-ANGLECOUNTSKETCH. We follow their notation that $O^*(f)$ denotes a function of the form $f \cdot \text{poly}(1/\epsilon, d, \log(m\Delta/\delta))$.

**Lemma 21** (RECTANGLECOUNTSKETCH, Tirthapura & Woodruff (2012)). *The data structure* RECTANGLECOUNTSKETCH($\gamma$) *can be updated rectangle-efficiently in time $O^*(\gamma^{-2})$. The total space is $O^*(\gamma^{-2})$ words. The data structure can be used to answer any query $i \in GF(\Delta)^d$, returning a number $\Phi(i)$ with $|\Phi(i) - x_i| \leq \gamma \|x\|_2$. The algorithm succeeds on all queries simultaneously with probability $\geq 1 - \delta$.*

Under an arbitrary mapping $g : GF(\Delta)^d \to \{1, 2, ..., \Delta\}$, they subsample the items into a logarithmic number $\phi = d \log(\Delta)$ of levels. In particular, for $j \in [\phi]$, $i \in D_j$ if $g(i) \leq \Delta^d/2^{j-1}$. In each level, they give an algorithm which invokes RECTANGLECOUNTSKETCH($\gamma$) to recover a list of the $F_p$ heavy hitters. Then they use the heavy hitters from each layer to calculate the final estimate.

**Lemma 22** (Tirthapura & Woodruff (2012)). *For $p > 2$, there is an algorithm which uses* RECTANGLECOUNTSKETCH($\gamma$) *with $\gamma = \Theta(\epsilon^{1+\frac{1}{p}}\alpha^{\frac{1}{p}}/\Delta^{d/2-d/p})$ and with probability at least $1 - \delta$ outputs a set $P$ of $O(1/\alpha)$ pairs $(i, x_i')$ such that $\sum_{(i,x_i')\in P} |x_i'|^p - |x_i|^p| \leq \epsilon \|x\|_p^p$ and all elements $i$ with $|x_i|^p \geq \alpha \|x\|_p^p$ appear as the first element of some pair in $P$. The algorithm uses $O^*(\gamma^{-2})$ words of space.*

Since the algorithm relies on a subroutine that finds the $\ell_p$-heavy hitters, the improvement with a heavy hitter oracle is similar to that for the $F_p$-moment problem. Specifically, suppose the heavy hitter oracle is able to indicate whether $|x_i|^p > \frac{1}{\Delta^{d/2}}\|x\|_p^p$. We know from Theorem 7 that we can use a rectangle-efficient estimator for the substream which contains all the heavy items, while for the light items, we can use the subsampling algorithm with rate $\Delta^{d/2}$. Note that for each substream, we have $\|x\|_0 \leq \Delta^{d/2}$ (for those $i$'s that are not in the substream, we see $x_i = 0$), so by Hölder's inequality, it holds that $\|x\|_2 \leq \Delta^{d/4-d/(2p)}\|x\|_p$. Following the proof of Lemma 22, we can set $\gamma = \Theta(\epsilon^{1+\frac{1}{p}}\alpha^{\frac{1}{p}}/\Delta^{d/4-d/(2p)})$ to find those heavy hitters. We thus have the following theorem.

**Theorem 23.** *Under the assumption of a heavy hitter oracle, there is a rectangle-efficient single-pass streaming algorithm which outputs a $(1 \pm \epsilon)$-approximation to $\|x\|_p^p$ with probability $1 - \delta$ for $p > 2$. It uses $O^*(\Delta^{d(1/2-1/p)})$ bits of space and $O^*(\Delta^{d(1/2-1/p)})$ time to process each rectangle in the stream.*

