# OpenReview forum: "Learning-Augmented Data Stream Algorithms"
_ICLR.cc/2020/Conference — Accept (Poster)_

### Official Review · AnonReviewer3 · 2019-10-22
**Official Blind Review #3**

**Rating:** 8

**Review:**

The paper talks about calculating various statistics over data streams. This is a very popular topic and is very relevant in big data analysis. A lot of work has been done in this general area and on the problems that are discussed in the paper. The new idea in the paper is better streaming algorithms under the assumption that there is a “heavy hitters” oracle that returns data items that have a lot of representation in the stream. The authors give provably better algorithms for the distinct elements problem, F_p moment problem (p > 2), and some more problems. These are important problems in streaming data analysis. They improve the space bounds and interestingly in some cases the bounds are better than what is possible without the oracle assumption. This also shows the power of such an oracle. There are experimental results to demonstrate the efficiency of the algorithms. At a high level the work seems good and interesting for a large audience interested in streaming data analysis. I have not gone over the proofs in detail (much of which is in the appendix).

- Even though oracle results are interesting, to make it practical it may make sense to talk about a more realistic, weaker oracle where some of the queries may be incorrect.
- It may even make sense to minimise the number of oracle calls which can be thought of as a resource and discuss the relationship between number of oracle calls and other resources such as space.


**Experience Assessment:**

I have read many papers in this area.

**Review Assessment: Checking Correctness Of Derivations And Theory:**

I assessed the sensibility of the derivations and theory.

**Review Assessment: Checking Correctness Of Experiments:**

I assessed the sensibility of the experiments.

**Review Assessment: Thoroughness In Paper Reading:**

I read the paper at least twice and used my best judgement in assessing the paper.

---

> ### Author Response · Authors · 2019-11-15
> **Response to Reviewer #3**
>
> We thank the reviewer for the comments on our paper.
> - We have included the result concerning a noisy oracle for the F_p moment estimation problem in the paper.
> - We like the question of minimizing the number of oracle calls. This is an interesting open problem and we intend to explore it in future work.

---

### Official Review · AnonReviewer2 · 2019-10-27
**Official Blind Review #2**

**Rating:** 8

**Review:**

Algorithms for Streaming data using a machine learning oracle is analyzed theoretically and empirically.

The idea is to build on some recent work (Hsu 19) which used RNNs to predict heavy hitters in streaming data. The purpose of this paper is to analyze whether such an oracle can help streaming algorithms to obtain improved bounds. I am not very familiar with this line of research so my comments will be more general in this case. The idea of improved bounds for streaming algorithms using machine learning oracle seems to be very appealing to me. The authors present novel theoretical results supporting this.

Experiments are performed on real as well as synthetic datasets using Hsu et al.’s method as an oracle.  Two real-world problems are selected, i.e., distinct packets in a network flow, Number of occurrences of each type of search query, and it is shown that using a oracle improves performance as compared to methods that do not use the oracle. Overall, I think the paper seems to be  an interesting direction which has both formal guarantees and experiments validating them in real-world datasets. One issue is perhaps, very little in terms of related work. I am not sure if this is the first work in this direction of proving bounds assuming an oracle or if there is some background work that the authors could provide to put this into context.

**Experience Assessment:**

I do not know much about this area.

**Review Assessment: Checking Correctness Of Derivations And Theory:**

I did not assess the derivations or theory.

**Review Assessment: Checking Correctness Of Experiments:**

I assessed the sensibility of the experiments.

**Review Assessment: Thoroughness In Paper Reading:**

I read the paper at least twice and used my best judgement in assessing the paper.

---

> ### Author Response · Authors · 2019-11-15
> **Response to Reviewer #2**
>
> We thank the reviewer for the comments on our paper. Designing more efficient streaming algorithms with machine learning techniques is a relatively new research topic and we have included more related work in our updated version of the manuscript (highlighted in the blue color).

---

### Official Review · AnonReviewer1 · 2019-11-05
**Official Blind Review #1**

**Rating:** 3

**Review:**

The paper presents algorithms for solving computational problems in a datastream model augmented with an oracle learned from data. The authors show that under this model, there exist algorithms that have significantly better time and space complexity than the current best known algorithms that do not use an oracle. The authors support their theoretical analysis with experiments in which the oracle is represented by a deep neural network and demonstrate improvement over classical algorithms that do not use machine learning.

Overall, this paper seems like a solid contribution to the literature. However, in its current state it does seem to be presented and motivated in a way that is appropriate for the audience of ML researchers at ICLR. It reads very much like a STOC theory paper, and a lot of the key ML details that would be relevant to audience at this conference seem to have been shoved under the rug in a way. Therefore my score for now is a weak reject, but I am very happy to increase the score if the authors address my presentations concerns.

Major comments:
* The oracle-augmented datasteam model needs to be contextualized better. I don't have a good sense of whether this is a reasonable theoretical model to explore and a lot of very basic questions remain unanswered for me. For example, how do I even know that the oracle in question exists? What are the particular assumptions under which it exists? What are the requirements on the training data, optimization ability, generalization error, etc. How do we know that we can create in practice ML learning models that are sufficiently accurate to serve as an oracle?

* The connections to deep learning seem arbitrary in some of the experiments. In one of the experiments, the authors train neural networks over a concatenation of IP address embeddings. Why do we need to use deep learning here? What is the benefit of using DL algorithms within the oracle-augmented datastream model? Is a simple algorithm enough? What algorithms should we ideally use in practice? What if you used simpler online learning algorithms with formal accuracy guarantees?

Minor comments:
* I thought there was a bit over-selling in intro. The authors say that they match the theoretical lower bounds for several problems. However, you are in a different computational model in which you now have access to an oracle. This needs to be made more explicitly, and language could be a bit toned down (e.g. in this model, we can obtain runtime that match or improve over lower bounds...)

**Experience Assessment:**

I do not know much about this area.

**Review Assessment: Checking Correctness Of Derivations And Theory:**

I did not assess the derivations or theory.

**Review Assessment: Checking Correctness Of Experiments:**

I assessed the sensibility of the experiments.

**Review Assessment: Thoroughness In Paper Reading:**

I read the paper at least twice and used my best judgement in assessing the paper.

---

> ### Author Response · Authors · 2019-11-15
> **Response to Reviewer #1**
>
> We thank the reviewer for the comments on our paper. A prior work of Hsu et al. (ICLR'18) showed that heavy hitter oracles exist and that they can be constructed using machine learning techniques. We are using the same type of oracles in our current submission. Similar oracles have been studied in previous works too, e.g., membership oracles for Bloom filters in Kraska et al. Both the previous works and the experiments in the current submission demonstrate that it is reasonable to make such an oracle assumption.
>
> We thank the reviewer for the questions on optimization ability, generalization error, etc. These are interesting research directions. The answers are very likely to depend on the application, data sets, etc., which we plan to study in the future.
>
> The prior work by Hsu et al. showed that the oracle trained by deep learning has high accuracy (see Section 5.3 in their paper): for Internet traffic data, the AUC score is 0.9, and for search query data, the AUC score is 0.8. The performance of a simple online algorithm would likely depend on the type of classifier used and input feature representation. Linear classifiers with IP addresses represented as individual bits are unlikely to work well because their expressive power is limited. For instance, at the very least, we would like our classifier to express a DNF hypothesis of the form:
> (IP address = a1) or (IP address  = a2) or ...
>
> We have updated the introduction to rephrase and clarify the lower bound claims. The added/modified text are highlighted in the blue color.

---

### Decision · Program_Chairs · 2019-12-19

**Decision:**

Accept (Poster)

**Comment:**

This paper theoretically analyzes the use of an oracle to predict various quantities in data stream models.  Building upon Hsu et al., (2019), the overriding goal is to examine the degree to which such an oracle is can provide memory and time improvements across broad streaming regimes.  In doing so, optimal bounds are derived in conjunction with a heavy hitter oracle.

Although the rebuttal and discussion period did not lead to a consensus in the scoring of this paper, two reviewers were highly supportive.  However, the primary criticism from the lone dissenting reviewer was based on the high-level presentation and motivation, and in particular, the impression that the paper read more like a STOC theory paper.  In this regard though, my belief is that the authors can easily tailor a revision to increase the accessibility to a wider ICLR audience.